# The Beneficial Effects of Astaxanthin on Glucose Metabolism and Modified Low-Density Lipoprotein in Healthy Volunteers and Subjects with Prediabetes

**DOI:** 10.3390/nu13124381

**Published:** 2021-12-07

**Authors:** Masaharu Urakaze, Chikaaki Kobashi, Yukihiro Satou, Kouichi Shigeta, Masahiro Toshima, Masatoshi Takagi, Jiro Takahashi, Hiroshi Nishida

**Affiliations:** 1Internal Medicine, Diabetes Center, Kamiichi General Hospital, Kamiichi-machi 930-0391, Toyama Prefecture, Japan; kobashi@kamiichi-hosp.jp; 2Internal Medicine, Kamiichi General Hospital, Kamiichi-machi 930-0391, Toyama Prefecture, Japan; sato@kamiichi-hosp.jp (Y.S.); sigeta@kamiichi-hosp.jp (K.S.); 3Vascular Surgery, Kamiichi General Hospital, Kamiichi-machi 930-0391, Toyama Prefecture, Japan; toshima@kamiichi-hosp.jp; 4Pharmacy, Kamiichi General Hospital, Kamiichi-machi 930-0391, Toyama Prefecture, Japan; dmcenter@kamiichi-hosp.jp; 5Fuji Chemical Industries Co., Ltd., Kamiichi-machi 930-0355, Toyama Prefecture, Japan; takahashi@fujichemical.co.jp (J.T.); h-nishida@fujichemical.co.jp (H.N.)

**Keywords:** astaxanthin, HbA1c, 75 g OGTT, malondialdehyde-modified low-density lipoprotein, apolipoprotein E atherosclerosis, prediabetes

## Abstract

Astaxanthin (ASTX) is an antioxidant agent. Recently, its use has been focused on the prevention of diabetes and atherosclerosis. We examined the effects of astaxanthin supplementation for 12 weeks on glucose metabolism, glycemic control, insulin sensitivity, lipid profiles and anthropometric indices in healthy volunteers including subjects with prediabetes with a randomized, placebo-controlled trial. Methods: We enrolled 53 subjects who met our inclusion criteria and administered them with 12 mg astaxanthin or a placebo once daily for 12 weeks. Subsequently, their HbA1c levels, lipid profiles and biochemical parameters were determined. The participants also underwent a 75 g oral glucose tolerance test (OGTT), vascular endothelial function test and measurement of the visceral fat area. Results: After astaxanthin supplementation for 12 weeks, glucose levels after 120 min in a 75 g OGTT significantly decreased compared to those before supplementation. Furthermore, the levels of HbA1c (5.64 ± 0.33 vs. 5.57 ± 0.39%, *p* < 0.05), apo E (4.43 ± 1.29 vs. 4.13 ± 1.24 mg/dL, *p* < 0.05) and malondialdehyde-modified low-density lipoprotein (87.3 ± 28.6 vs. 76.3 ± 24.6 U/L, *p* < 0.05) were also reduced, whereas total cholesterol (TC), triglyceride (TG) and high-density lipoprotein-C (HDL-C) levels were unaltered. The Matuda index, which is one of the parameters of insulin resistance, was improved in the ASTX group compared to that before supplementation. Conclusions: our results suggest that ASTX may have preventive effects against diabetes and atherosclerosis and may be a novel complementary treatment option for the prevention of diabetes in healthy volunteers, including subjects with prediabetes, without adverse effects.

## 1. Introduction

Type 2 diabetes mellitus describes a group of metabolic disorders characterized by high blood glucose levels due to an imbalance between insulin supply and demand. Patients with diabetes often have dyslipidemia and an increased risk of developing a number of life-threatening disorders, such as myocardial infarction and stroke, resulting in higher medical care costs, a reduced quality of life and increased mortality [1].

The global prevalence of diabetes and impaired glucose tolerance in adults has been increasing in recent decades. In many countries and regions, the number of people with impaired glucose tolerance, including diabetes, has been boosted by rapid urbanization and dramatic changes leading to a sedentary lifestyle [2]. The International Diabetes Federation (IDF) estimates that 425 million individuals globally, or 8.8% of the world’s population (1 in 11 adults), have diabetes, with 629 million adults, or 9.9% of the world’s population, expected to develop diabetes by 2045. In addition, 7.3% of the world’s population, or 352 million individuals, have impaired glucose tolerance (IGT) and are considered to be increased risk of developing diabetes, with an expectation that this will increase to 532 million, or 8.3%, in 2045 [3].

The prevention of type 2 diabetes is an urgent issue to maintain a person’s quality of life and increase their healthy life-span. The major pathogenic factor of type 2 diabetes is insulin resistance; thus, the amelioration of insulin resistance is essential for preventing type 2 diabetes. Hurrle and Hsu [4] reported that oxidative stress has been recognized as a key mechanism in insulin resistance, and there is a strong correlation between the state of oxidative stress in the body and the incidence of insulin resistance and even late-stage diabetes cases, which is linked to the therapeutic potential of antioxidants.

Among many antioxidants, astaxanthin (ASTX) is one of the most powerful compounds and has been approved as a dietary supplement by the FDA in the USA. ASTX is a red xanthophyll carotenoid (3,3′-dihydroxy-β,β-carotene−4,4′-dione), which is a fat-soluble red pigment found in several species, including salmon, shrimp, crustacean, microalgae such as *Haematococcus pluvialis* and birds [5]. A major dietary source of ASTX is *Haematococcus pluvialis* (*H. pluvialis*), a green microalga that has a high ASTX content and is frequently ingested by fish and other marine organisms.

Miki [6] examined the quenching or scavenging effects of animal carotenoids against active oxygen species, singlet oxygen and hydroxy-radical, and against organic free radicals and reported that the potencies of astaxanthin are approximately 10 times stronger than those of other carotenoids (zeaxanthin, lutein, tunaxanthin, canthaxanthin and beta-carotene) and 100 times greater than those of alpha tocopherol. Nishida et al. [7] reported that astaxanthin exhibited the most potent singlet oxygen-quenching activity among the compounds tested (lutein, α-lipoic acid, ubiquinone-10(CoQ10), caffeic acid, quercetin, resveratrol, gallic acid, pyrocatechol, pyrogallol, BHT, sesamin, L(+)-ascorbic acid, α-tocopherol, probucol, canthaxanthin, lycopene, β-cryptoxanthin, Trolox, edaravon, curcumin, epigallocatechin gallate, capsaicin, β-carotene and fucoxanthin). ASTX inhibits lipid peroxidation 100 to 500-fold more strongly than vitamin E in vitro [8] and has a several-fold greater free radical antioxidant potency than vitamin E and β-carotene [9]. ASTX has a hydroxyl radical scavenging capability even in aqueous solution [10].

Many studies have documented that the dietary consumption of ASX can prevent or reduce the risk of various medical conditions in humans and animals [11,12,13,14,15]. Several investigators have already reported that ASTX positively alters cholesterol and lipid metabolism in healthy humans or those with cardiovascular diseases who have enhanced antioxidant properties and that these effects are mediated through antioxidant defense mechanisms [12,16]. However, to the best of our knowledge, we have no reports of human studies regarding the prevention of diabetes by the supplementation of ASTX in high-risk subjects, such as those with prediabetes with increased tissue oxidative stress and inflammation.

Therefore, we conducted a 12-week trial to assess the effects of 12 mg of daily oral ASTX supplementation on glucose metabolism (using the 75 g oral glucose tolerance test (75 g OGTT)), glycemic control, insulin sensitivity, lipid profiles and anthropometric indices in healthy volunteers and subjects with prediabetes.

## 2. Participants and Methods

### 2.1. Study Population

The present study was carried out in accordance with the CONSORT guidelines, and a participant flow chart is presented in Figure 1. This study was a single-center, double-blind, randomized, parallel, placebo-controlled trial. Fifty-three healthy subjects were recruited at the Checkup Center at the Kamiichi General Hospital. The subjects who met our inclusion criteria were then enrolled after written consent was obtained. Subjects were considered eligible if they met the following criteria: an age of 20–74 years; fasting blood glucose under 125 mg/dL; no pregnancy or lactation; the absence of self-reported specific diseases and malignancies, kidney failure, heart disease, thyroid disorder and other inflammatory diseases; finally, patients also did not report taking any supplements during the 6 months prior to recruitment. Subjects were excluded if they were taking supplements or medicines for other diseases, were unable to follow the study or had participated in another clinical drug study within 30 days of the present study. Subjects with prediabetes were considered eligible for the study. Participants were then allocated to one of two groups using a prepared randomization system. Randomization was performed by an assistant using the block randomization method, and stratified randomization was employed to match participants based on gender distribution. As such, an equal number of participants were assigned to receive either ASTX or a placebo. The intervention allocation was blinded for both investigators and participants. During the course of the study, participants were advised to maintain their current lifestyle. The protocol of this randomized clinical trial was approved by the Research Ethics Committee of the Japan Society of Nutrition and Food Science (IRB: H27–52), and was registered in UMIN-CTR with UMIN000045518. All participants provided their written informed consent.

### 2.2. Study Protocol

Each participant was required to ingest one tablet of either the placebo or the ASTX supplement daily for 12 weeks. The inactive ingredient content in both tablets was the same: 112.8 mg of mid-chain fatty acids and 7.2 mg of mixed tocopherols. The active tablet contained 12 mg of ASTX (Fuji Chemical Industries Co. Ltd., Kamiichi-machi, Toyama-ken, Japan), which was derived from *H. pluvialis*. The daily ingestion amount of 12 mg and ingestion period of 12 weeks were set based on previous reports [17].

Blood samples were collected before and after 12 weeks of supplementation with ASTX or placebo after overnight fasting (10–12 h). The plasma concentration of ASTX was thus determined within 20 h of the last pill, coinciding with the plasma elimination half-time (T1/2) of ASTX, which is approximately 16 h after oral administration [18].

At each 4-week visit, participants’ adherence to the study protocol was assessed by counting the remaining tablets. Participants were also asked to maintain their habitual diet, lifestyle and tablet ingestion.

### 2.3. Sample Processing and Data Collection

The medical staff carried out all of the data collection and measurements. The interviewers collected several demographic characteristics, including age, sex and medical history. The body mass index was calculated as weight in kg divided by height in meters for each participant, and the abdominal circumferences were measured on a horizontal plane at the level of the umbilicus.

The visceral fat area (VFA) at the level of the umbilicus was estimated by using a DUALSCAN HDS-2000 (OMRON Healthcare Co., Kyoto, Japan).

Body composition, including skeletal muscle mass and body fat mass, was evaluated using the InBody 720 (InBody Co., Ltd., Seoul, Korea).

Vascular endothelial function was examined using Endo-PAT 2000 (Itamar Medical Ltd., Caesarea, Israel). The subjects remained quiet and relaxed and sat in a comfortable position during the strictly performed testing protocol. First, the subjects placed their index finger on the Endo-PAT biosensor probe, and vascular endothelial function was detected on one side. The other side represented the control, which responded to dynamic changes in whole blood vessels. A standard cuff (Hokanson AG101, D. E. Hokanson Inc., Bellevue, WA, USA) was placed on the upper 2 cm of the brachial artery without pneumatic compression. Second, a 1 min signal and stability test was performed, and the baseline tension data were collected for 5 min using a fast-filling gas at the end of the cuff (a general pressure of 200 mmHg or higher shrinkage pressure of 60 mmHg). The cuff was quickly deflated, and blood flow signal acquisition was processed for 5 min. Finally, Endo-PAT 2000 computer software (Medical Itamar) was used to automatically collect the signal data in order to obtain the reaction of the vascular endothelial function of the vascular reactive hyperemia index (RHI) [19,20].

At each visit, blood pressure was measured using a mercury sphygmomanometer after 15 min of rest, which consisted of sitting in a stress-free condition. An oral glucose tolerance test was performed in a fasting state. Venous blood samples were collected for biochemical measurements before and after supplementation for 12 weeks. Collected blood samples were centrifuged, and the isolated serum was stored at −80 °C. Blood glucose, lipid profiles and other biochemical parameters were measured using an auto-analyzer based on the colorimetric method. Plasma concentrations of ASTX were determined using reverse-phase high-performance liquid chromatography (Alliance 2690, Waters, Milford, MA, USA), where transb-apo-8′-carotenal (Sigma Chem. Co., St. Louis, MO, USA) was used as an internal standard. EDTA was used as anticoagulant for the plasma samples. Glycolysis inhibitor, sodium fluoride, was used for glucose testing.

### 2.4. Statistical Analysis

All data are presented as the mean ± standard deviation for continuous variables, and as numbers (percentages) for categorical variables. The Wilcoxon signed-rank test was used to test for significance in the comparisons within each group. The Mann–Whitney U test was used to test for significant differences between groups. All statistical analyses were performed using SPSS (Version 20, SPSS Inc., Chicago, IL, USA), and significance was set at *p* < 0.05.

## 3. Results

### 3.1. Baseline Anthropometric and Demographic Characteristics of the Participants

Fifty-three patients were enrolled in the study, and 44 of them completed the study. A summary of their physical characteristics and the medication received is presented in Table 1. Notably, the baseline data did not significantly differ between the ASTX and placebo groups. By counting the recollected capsule boxes (either empty or those containing ASTX supplements that were not consumed) at every 4-week visit, we confirmed that the compliance of our participants was satisfactory.

### 3.2. Changes in the Plasma ASTX Concentration

In the placebo group, the plasma ASTX concentration was undetectable in all the participants at weeks 0, 4, 8 and 12. However, in the ASTX group, the plasma ASTX concentration was undetectable at baseline and increased to 122.69 ng/mL a month later; this level was maintained until 3 months later (Figure 2).

### 3.3. Changes in the Glucose Metabolic Parameters

After ASTX (12 mg/day) supplementation for 12 weeks, the glucose level at 120 min, determined by 75 g OGTT, was significantly decreased compared to that at baseline, as shown by the Wilcoxon signed-rank test (Figure 3). The insulin level at 120 min, determined by 75 g OGTT, was also significantly decreased (39.62 ± 34.3 to 19.09 ± 11.9 μU/mL, *p* < 0.01) after ASTX (12 mg/day) supplementation. Matsuda index was also significantly improved (8.29 ± 3.96 vs. 11.45 ± 4.81, *p* < 0.01) after ASTX (12 mg/day) supplementation. The levels of HbA1c were significantly decreased compared to that at baseline (5.64 ± 0.33 vs. 5.57 ± 0.39%, * *p* < 0.05, Figure 4a), as shown by the Wilcoxon signed-rank test. This is the first report of this in a human study. The levels of HbA1c in prediabetic subjects, ranging from 5.6 to 6.4, were also significantly decreased compared to those at baseline, as shown by the Wilcoxon signed-rank test (5.82 ± 0.22 vs. 5.73 ± 0.35%, * *p* < 0.05, Figure 4b). The changes in the level of HbA1c between the astaxanthin and placebo groups was not significant, as shown by the Mann–Whitney U test.

### 3.4. Changes in the Lipid Parameters

The levels of apolipoprotein E (4.43 ± 1.29 vs. 4.13 ± 1.24 mg/dL, *p* < 0.05, Figure 5) and malondialdehyde-modified low-density lipoprotein (MDA-LDL) (87.3 ± 28.6 vs. 76.3 ± 24.6 U/L, *p* < 0.05, Figure 6) were significantly reduced, whereas the levels of triglycerides (TGs), total cholesterol (TC) and HDL-C were not affected significantly (Table 2). 

### 3.5. Changes in Other Parameters

The RHI, an index of endothelial function, was improved (1.84 ± 0.36 vs. 2.12 ± 0.55, *p* < 0.05) after ASTX (12 mg/day) supplementation. None of the other parameters was significantly altered.

## 4. Discussion

To the best of our knowledge, this is the first report showing that daily oral administration of 12 mg ASTX, which is a powerful antioxidant, significantly reduces the levels of HbA1c, a major indicator of blood glucose control in the human study, suggesting that ASTX may have a preventive effect on diabetes in healthy volunteers, including subjects with prediabetes. Ursoniu et al. [21] performed a meta-analysis to evaluate the efficacy of astaxanthin supplementation on plasma lipid and glucose concentrations and reported that a slight glucose-lowering effect, not HbA1c, was observed.

Multiple studies have reported the antidiabetic effects of ASTX. In animal studies, Uchiyama et al. [22] reported that ASTX supplementation protected pancreatic beta-cells against glucose toxicity by decreasing the level of blood glucose and oxidative stress induced by hyperglycemia in diabetic db/db mice. ASTX may reduce hyperglycemia and improve insulin resistance and insulin secretion by the improvement of glucose dysmetabolism and beta-cell dysfunction via GLUT4 regulation [13,23]. In human studies, Mashhadi et al. [24] reported that the 8-week administration of ASTX (8 mg/day) supplementation reduced the fructosamine concentration (*p* < 0.05) and marginally reduced the fasting plasma glucose concentration in diabetic patients.

In our study, ASTX significantly reduced the glucose level at 120 min as determined by 75 g OGTT, which was concomitant with a decrease in serum immunoreactive insulin concentration relative to baseline. These data suggest that insulin sensitivity is improved by supplementation with ASTX. We also calculated the index of insulin resistance (Matsuda index) from the data of the 75 g OGTT, which showed that the index improved with ASTX administration.

Concerning the lipid profile, the affirmative effects of ASTX on healthy volunteers remain unclear. Ursoniu et al. [21] reported by a meta-analysis that no significant effect of supplementation with astaxanthin on the plasma lipid profile was observed. Iwamoto et al. [25] reported that with the supplementation of ASTX, serum TC, TG, LDL-C, HDL-C and apolipoprotein levels did not change significantly, and no difference in the oxidation of LDL was observed. On the other hand, in our study, we observed that the levels of apo E and MDA-LDL—MDA-LDL is a major atherogenic factor—were markedly decreased in the ASTX group compared to the control group, suggesting that ASTX may have beneficial effects on the prevention of atherosclerosis in patients with prediabetes and diabetes, whereas TC, TG, LDL-C and HDL-C were not significantly changed. Yoshida et al. [14] reported that the administration of 12 mg of ASTX significantly increased serum HDL-cholesterol in subjects with mild hyperlipidemia. In our results, we did not find an increase in the serum level of HDL-cholesterol in both ASTX and placebo groups, which was probably due to the difference between healthy participants and dyslipidemic patients.

Concerning blood pressure, a decrease in blood pressure was also reported in patients receiving ASTX with type 2 diabetes mellitus [24]. However, in our results, we did not find a decrease in blood pressure in both the ASTX and placebo groups, which was probably due to the difference between healthy participants and hypertensive patients.

Concerning endothelial function, there have been no reports to date of human studies showing the endothelial effects of ASTX on endothelial function. In animal studies, Wang et al. [26] reported that ASTX could inhibit homocysteine-induced endothelial dysfunction by suppressing the homocysteine-induced activation of the VEGF–VEGFR2–FAK signaling axis in vitro. Zhao et al. [27] reported that ASTX treatment could significantly decrease serum oxLDL and aortic MDA levels, attenuate blunted endothelium-dependent vasodilator responses to ACh, upregulate eNOS expression and decrease LOX-1 expression. In our human study, for the first time, we also show that ASTX improves endothelial function in humans.

Our present study has some limitations that should be addressed. First, although we reported that ASTX supplementation improved insulin sensitivity, we did not carry out experiments to determine the molecular mechanisms underlying the effects of ASTX in human cells to support our findings, and this should be addressed in future studies. Second, we could not compare the hypoglycemic effects of ASTX with those of diabetic drugs on the market, as ASTX has not yet been formulated as a clinical drug with exact recommended dosages. However, these findings may provide new insights into the potential role of ASTX in modulating several conditions in individuals with type 2 diabetes mellitus.

## 5. Conclusions

We demonstrated that ASTX decreases the level of HbA1c in healthy volunteers and subjects with prediabetes. Our results suggest that ASTX may play a role in glucose control, and that might be clinically beneficial for the prevention of type 2 diabetes in subjects with or without prediabetes. Further research is needed to clarify the effect of ASTX on the prevention of type 2 diabetes. Furthermore, our results indicate that ASTX may be a novel complementary treatment with potential impacts on diabetes and diabetic complications without adverse effects. These beneficial effects should be tested in type 2 diabetes situation.

## Figures and Tables

**Figure 1 nutrients-13-04381-f001:**
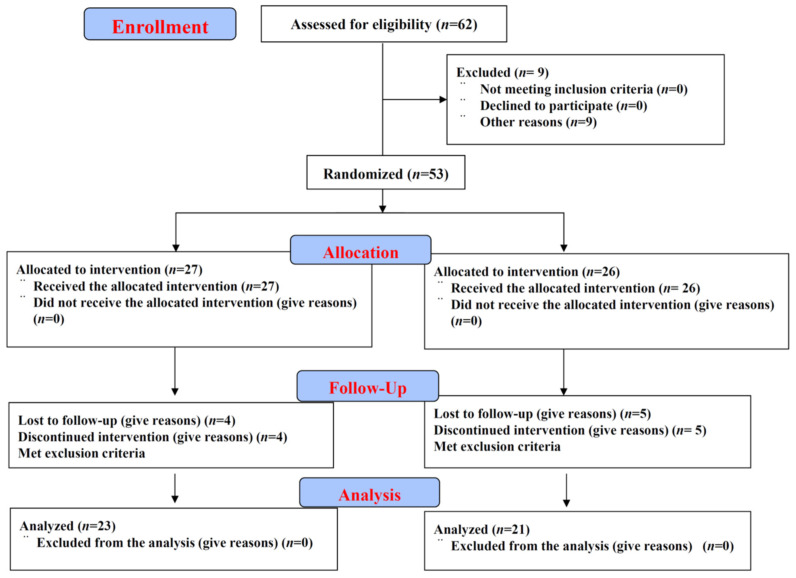
Flow diagram of the progress through the phases of a randomized trial of two groups (enrolment, intervention allocation, follow-up and data analysis).

**Figure 2 nutrients-13-04381-f002:**
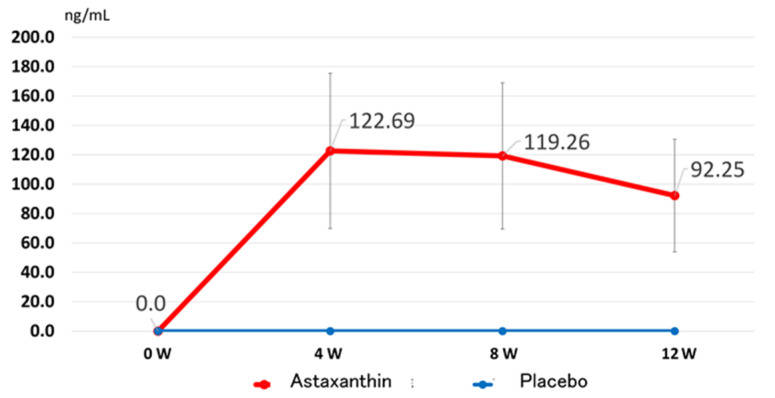
Changes in the plasma concentration of astaxanthin (ASTX) after supplementation with 12 mg ASTX for 4, 8 and 12 weeks. All data are presented as the mean ± standard deviation.

**Figure 3 nutrients-13-04381-f003:**
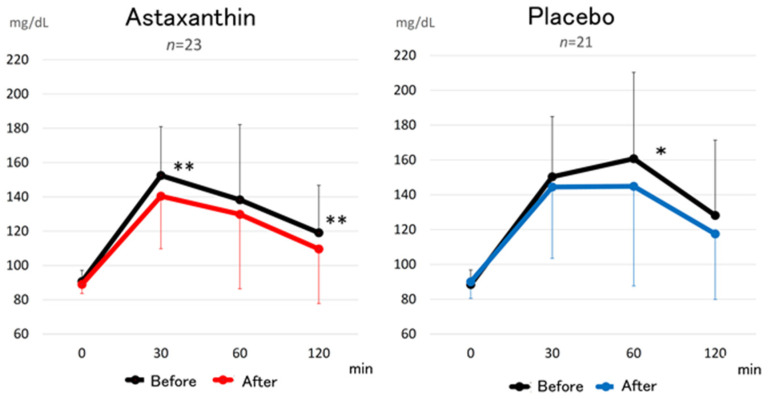
Changes in the plasma glucose concentration based on 75 g OGTT before and after supplementation with 12 mg ASTX for 12 weeks. Figure 4a shows the results in all subjects. Figure 4b shows the results in subjects with prediabetes, whose HbA1c range was from 5.6 to 6.4. All data are presented as the mean ± standard deviation. The Wilcoxon signed-rank test was used to test for significant differences within each group. *: *p* < 0.05, **: *p* < 0.01.

**Figure 4 nutrients-13-04381-f004:**
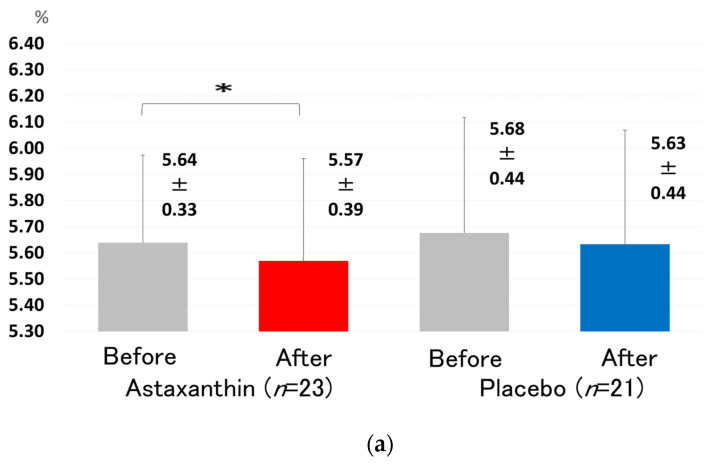
Changes in the levels of HbA1c before and after supplementation with 12 mg ASTX for 12 weeks. (**a**) All subjects. (**b**) Subjects with prediabetes with a HbA1c range from 5.6 to 6.4. All data are presented as the mean ± standard deviation. The Wilcoxon signed-rank test was used to test for significant differences within each group. *: *p* < 0.05.

**Figure 5 nutrients-13-04381-f005:**
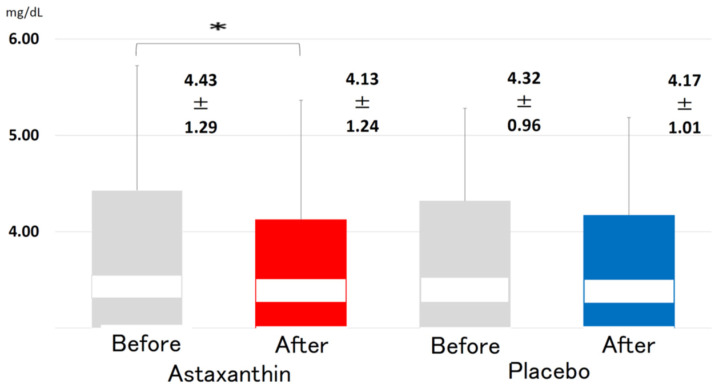
Changes in the serum levels of apo-E before and after supplementation with 12 mg ASTX for 12 weeks. All data are presented as the mean ± standard deviation. The Wilcoxon signed-rank test was used to test for significant differences within each group. *: *p* < 0.05.

**Figure 6 nutrients-13-04381-f006:**
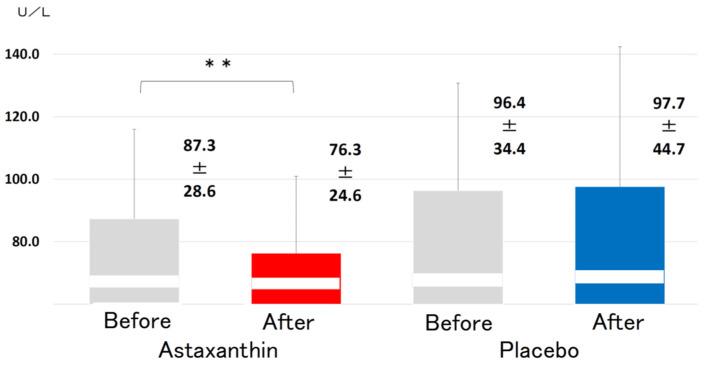
Changes in the serum levels of malondialdehyde-modified low-density lipoprotein (MDA-LDL) before and after supplementation with 12 mg ASTX for 12 weeks. All data are presented as the mean ± standard deviation. The Wilcoxon signed-rank test was used to test for significant differences within each group. **: *p* < 0.01.

**Table 1 nutrients-13-04381-t001:** Baseline characteristics of the participants.

	Astaxanthin *n* = 23 (Male: 8 Female: 15)	Placebo *n* = 21 (Male: 7 Female: 14)	
Age (years)	46.2 ± 13.7	48.2 ± 12.0	n.s.
BMI	21.0 ± 2.0	23.9 ± 5.4	n.s.
Systolic BP (mmHg)	110.1 ± 18.9	117.9 ± 15.6	n.s.
Diastolic BP (mmHg)	68.3 ± 13.8	71.8 ± 9.6	n.s.
Pulse Rate	71.1 ± 7.7	72.6 ± 9.6	n.s.
Waist Circumference (cm)	77.1 ± 7.4	85.3 ± 12.6	*
Subjects with prediabetes (*n*) (HbA1c 5.6~6.4)	16	13	

BMI: body mass index, BP: blood pressure, n.s.; not significant, * *p* < 0.05. The Mann–Whitney U test was used to test for significant differences between the groups.

**Table 2 nutrients-13-04381-t002:** Changes in lipids profile after supplementation with 12 mg ASTX for 12 weeks.

	Astaxanthin (*n* = 23)				Placebo (*n* = 21)			
	Before		After		Before		After	
	Mean	SD	Mean	SD	Mean	SD	Mean	SD
TC (mg/dL)	199.4	25.8	200.3	30.4	213.9	44.5	214.5	40.3
TG (mg/dL)	84.9	39.0	90.3	53.7	95.3	51.8	108.8	77.5
HDL-C (mg/dL)	64.0	14.3	64.7	15.3	59.4	12.3	57.0	11.3
LDL-C (mg/dL) (Friedewalde)	118.5	22.1	117.5	24.2	135.5	37.4	135.8	36.4
non HDL-C (mg/dL)	135.5	23.7	135.6	26.6	154.4	42.8	157.5	41.7
MDA-LDL (U/L)	87.3	28.6	76.3 **	24.6	96.4	34.4	97.7	44.7

The Wilcoxon signed-rank test was used to test for significance in comparisons within each group. **: *p* < 0.01.

## Data Availability

This study was registered in UMIN-CTR with UMIN000045518.

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
