# Peer review of "The Beneficial Effects of Astaxanthin on Glucose Metabolism and Modified Low-Density Lipoprotein in Healthy Volunteers and Subjects with Prediabetes"

_nutrients, 2021, doi:10.3390/nu13124381_

Round 1

Reviewer 1 Report

The Authors have evaluated the impact of astaxanthin on glucose and lipid metabolism. Although this topic is innovate and potentially interesting for the readers, some comments must be addressed.

1) Study population: The study was performed on a small group of subjects - out of 53, only 44 completed the study. The age of the participants varied widely (20-74 yo) which could have  a significant impact on the results.

2) Study protocol: The authors didn't provide an information wheter the study was a single, double or triple-blind study.

3) Sample processing and data collection: The authors didn't clearly define what type of samples was collected and tested - in the manuscript we can find different terms like: "plasma ASTX", "serum samples" etc. What type of anticoagulant was used for the plasma samples? Has whole blood samples been drawn for HbA1c testing? Moreover, the Authors didn't describe in detail what methods and analyzers were used for biochemical tests.  Has the method for HbA1c determination been certified by the IFCC or NGSP? This part should be thoroughly corrected.

4) Statistical analysis: Have the Authors checked the distribution of all variables? The results are presented as  mean+/- SD, therefore their distribution should be normal. If not, the results should be presented as median and 25th-75th percentile and other analyzes should be performed using non-parametric tests.

5) The results: Tha authors haven't presented the potential impact of age on study results. Although the Authors have emphasized that ASTX reduces HbA1c levels, which is a very  promising result, the follow-up time (12 weeks)  seems to be relatively short, considering the fact that HbA1c reflects the average glycaemia over the last 3 months.

Author Response

We provide a point by point response to the reviewers comments. Please find the attached file.  

Reviewer 2 Report

This manuscript is well written and the main ideas and objectives are clearly stated. this reviewer acknowledge the quality of the methods description and the presentation of the results. However, some conclusions appear without supporting results. for example, in line 254 on the discussion section it is stated that insulin levels were decreased with ASTX treatment but this was not shown in the results or even referred to be not shown or in supplementary data. the same for the Matsuda Index referred in the following sentence.

also, in line 285 the authors say that ASTX improves endothelial function but in fact nothing is presented that can lead to this conclusions. 

finally, in line 267 the authors make conclusions on the ASTX treatment in diabetic patients but they have to be careful when they make this statement since the presented results are only in control and pre-diabetic patients. in agreement with this, at the end of the conclusion another limitation and future perspective should be included saying that these beneficial effects should be tested in type 2 diabetes situation.

Author Response

We provide a point-by-point response to reviewers comments. Please find the attached file.  

Round 2

Reviewer 1 Report

In the Cover letter, the Authors stated that they had improved the information on blood samples and test methodology. However, several errors still appear in the revised version of the manuscript, for example:

  • line 160: "isolated serum" instead of "isolated plasma";
  • line 162: "based on the colorimetric method", without providing information about the type of autoanalyzer;
  • there is no information whether plasma with glycolysis inhibitors (e.g. sodium fluoride) was used for glucose and OGGT testing (the Authors added only: "EDTA was used as anticoagulant for the plasma samples", line 165).

Referring to comments on the influence of age on research results, the Authors replied: "The age of the main participants from 30s to 60s. It was hard to present the potential impact of age on study results". This explanation is not sufficient. The Authors should check whether there are differences in outcomes between the group, e.g. under and over 40 or 50 years of age. Thus, it would be confirmed whether ASTX supplementation brings similar beneficial effects in younger as well as in older subjects, who usually have a worse metabolic status and where prediabetes is more frequent. I leave the decision to take these results into account to both the Authors and the Editor.

Author Response

Reviewer 1

In the Cover letter, the Authors stated that they had improved the information on blood samples and test methodology. However, several errors still appear in the revised version of the manuscript, for example:

line 160: "isolated serum" instead of "isolated plasma";

Reply; isolated plasma was not stored at −80 °C.   

line 162: "based on the colorimetric method", without providing information about the type of autoanalyzer;

there is no information whether plasma with glycolysis inhibitors (e.g. sodium fluoride) was used for glucose and OGGT testing (the Authors added only: "EDTA was used as anticoagulant for the plasma samples", line 165).

Reply; We added the statement in Sample Processing and Data Collection.

Glycolysis inhibitor, sodium fluoride, was used for glucose testing.

Referring to comments on the influence of age on research results, the Authors replied: "The age of the main participants from 30s to 60s. It was hard to present the potential impact of age on study results". This explanation is not sufficient. The Authors should check whether there are differences in outcomes between the group, e.g. under and over 40 or 50 years of age. Thus, it would be confirmed whether ASTX supplementation brings similar beneficial effects in younger as well as in older subjects, who usually have a worse metabolic status and where prediabetes is more frequent. I leave the decision to take these results into account to both the Authors and the Editor.

Reply; Your comment is very important. We checked the differences in HbA1c  between the group under and over 45 years of age. However,there was no differences between the two groups. It might be due to small number (under 45 years 12 subjects, over 45 years 11 subjects).  So, it was hard to present the potential impact of age on our study results.

Reviewer 2 Report

I acknowledge the efforts of the authors to follow the suggestions and since all of them where correctly followed I support the publication of this manuscript in the present form.

Author Response

I acknowledge the efforts of the authors to follow the suggestions and since all of them where correctly followed I support the publication of this manuscript in the present form.

Thank you for your comments.

This manuscript is a resubmission of an earlier submission. The following is a list of the peer review reports and author responses from that submission.

Round 1

Reviewer 1 Report

Please prepare the manuscript according to the journals’ instructions for authors.
The abstract should be improved.
Please explain all abbreviations used in the abstract and main text.
A citation should be inserted directly after the author's name, e.g., Nishida et al. [7].
The key question should be changed. In the study, the authors did not assess the effect of dietary intake of astaxanthin but the effect of astaxanthin supplementation.
The section ”How might this impact clinical practice in the foreseeable future?” should be also improved. The authors did not assess the effect of astaxanthin on the risk of diabetes or atherosclerosis.
I suggest changing „healthy volunteers, including subjects with prediabetes” to „healthy volunteers and subjects with prediabetes” or „subjects with impaired fasting glucose”.
The following sentence ,,A major dietary source of ASTX is Haematococcus pluvialis (H. pluvialis), a green microalga that has a high ASTX content, and is frequently ingested by fish and other marine organisms.” should be inserted after the following sentence ”ASTX is a red xanthophyll carotenoid (3,3′-83 dihydroxy-β,β-carotene-4,4′-dione), which is a fat-soluble red pigment found in several species, including salmon, shrimp, crustaceans, and microalgae such as Haematococcus pluvialis, and birds 85 [5].”.
Please cite original studies (not review) here: „Many studies have documented that dietary consumption of ASX can prevent or reduce the risk of various medical conditions in humans and animals [11].”.
,,However, to the best of our knowledge, we have no reports about the prevention of diabetes in high-risk people, such as those with prediabetes with increased tissue oxidative stress and inflammation by the dietary intake of ASTX.” This sentence is not related to the study and needs to be correct.
Please provide the CONSORT checklist in the supplementary file.
Please explain how did you perform the randomisation?
Where did the authors register the study protocol?
Please provide more details about the methods which you used.
Please check if the correct study is cited here: „Finally, Endo-PAT 2000 computer 164 software (Medical Itamar) was used to automatically collect signal data in order to obtain the reaction of vascular endothelial function of the vascular reactive hyperemia index (RHI) [15].”.
Did the authors assess the dietary habits of the study population?
Did the authors calculate the minimal sample size?
In the Statistical analysis section please add the information on how you check the normal distribution of parameters?
Please compare the effect of astaxanthin supplementation in normal and impaired fasting glucose subjects.
The authors should also compare the effect of astaxanthin supplementation between the astaxanthin and placebo groups.
Are you sure that in Figure 1 the number of subjects who received the intervention should be 0?
„Participants were also asked to maintain their habitual diet, lifestyle, and tablet ingestion.” In the Study population section, you mentioned that participants were advised to maintain a healthy lifestyle.
Please add to Table 1 information about how many participants had prediabetes in both groups.
The authors should add information about the difference between the groups at baseline.
The sentence should be check if it is correct: „The levels of HbA1c 202 (5.64±0.33 vs 5.57±0.39ï¼…, p < 0.05, Fig 4), which is the first report in the human study.”.
Please prepare Table 2 according to journal recommendation.
Please explain how recommendations to provide a healthy lifestyle had an impact on the study results.
Please explain why the authors choose a dose 12 mg/d of astaxanthin?
Please check citations here: „In animal studies, several investigators reported that ASTX supplementation protected pancreatic beta-cells against glucose toxicity by decreasing the level of blood glucose and oxidative stress induced by hyperglycemia in diabetic db/db mice [12, 13, 16].”
Please check the citation in the following sentence: „In concern with blood pressure, a decrease in blood pressure was also reported in patients receiving ASTX with type 2 diabetes mellitus [21]." Te cited study was conducted in hypertensive rats.
Please cite in the discussion the results of the meta-analysis which assessed the effect of astaxanthin supplementation on lipid profile and glucose levels.
What are the strengths and novelty of the study?
Please add the conclusions section to the paper.

Author Response

Comments and Suggestions for Authors of Reviewer 1

  1. Please prepare the manuscript according to the journals’ instructions for authors.

1Reply: Our revised manuscript is now checked by MDPI English Editing Services.

  1. The abstract should be improved.

2Reply: Our revised manuscript is now checked by MDPI English Editing Services.

  1. Please explain all abbreviations used in the abstract and main text.

3Reply: Our revised manuscript is now checked by MDPI English Editing Services.

  1. A citation should be inserted directly after the author's name, e.g., Nishida et al. [7].

4Reply: We did it.  Our revised manuscript is now checked by MDPI English Editing Services.

  1. The key question should be changed. In the study, the authors did not assess the effect of dietary intake of astaxanthin but the effect of astaxanthin supplementation.

5Reply: The key question was changed to [Does the supplementation of ASTX have beneficial effects on glucose metabolism in healthy volunteers and subjects with prediabetes? ]  However, in our revised manuscript, parts of belows were deleted according to journal style. 

What is the key question?

Does the dietary intake of ASTX have beneficial effects on glucose metabolism in pre-diabetic subjects?

What are the new findings?

The glucose level at 120 min as determined by a 75 g oral glucose tolerance test was significantly decreased relative to baseline values following ASTX supplementation.

The levels of HbA1c was reduced by ASTX supplementation. 

Malondialdehyde-modified low-density lipoprotein levels were reduced by ASTX supplementation.

How might this impact clinical practice in the foreseeable future?

ASTX may have preventive effects against diabetes and atherosclerosis and may be a novel complementary treatment option for the prevention of diabetes in individuals with prediabetes without adverse effects.

  1. The section ”How might this impact clinical practice in the foreseeable future?” should be also improved. The authors did not assess the effect of astaxanthin on the risk of diabetes or atherosclerosis.

6Reply: In our revised manuscript, parts of the section was deleted according to journal style.

  1. I suggest changing „healthy volunteers, including subjects with prediabetes” to „healthy volunteers and subjects with prediabetes” or „subjects with impaired fasting glucose”.

7Reply: We changed [healthy volunteers, including subjects with prediabetes] to [healthy volunteers and subjects with prediabetes].

  1. The following sentence ,,A major dietary source of ASTX is Haematococcus pluvialis (H. pluvialis), a green microalga that has a high ASTX content, and is frequently ingested by fish and other marine organisms.” should be inserted after the following sentence ASTX is a red xanthophyll carotenoid (3,3-83 dihydroxy-β,β-carotene-4,4-dione), which is a fat-soluble red pigment found in several species, including salmon, shrimp, crustaceans, and microalgae such as Haematococcus pluvialis, and birds 85 [5]..

8Reply: It was changed in our revised manuscript as your comment. 

  1. Please cite original studies (not review) here: „Many studies have documented that dietary consumption of ASX can prevent or reduce the risk of various medical conditions in humans and animals [11].”.

9Reply: We cite original studies as follows.

12.Kimura M, Iida M, Yamauchi H, et al. Astaxanthin supplementation effects on adipocyte size and lipid profile in OLETF rats with hyperphagia and visceral fat accumulation. J Funct Foods 2014; 11: 114–120.

13.Bhuvaneswari S and Anuradha CV. Astaxanthin prevents loss of insulin signaling and improves glucose metabolism in liver of insulin resistant mice. Can J Physiol Pharmacol 2012; 90:1544–1552

14.Yoshida H, Yanai H, Ito K, et al. Administration of natural astaxanthin increases serum HDL-cholesterol and adiponectin in subjects with mild hyperlipidemia. Atherosclerosis 2010; 209:520–523.

15.Monroy-Ruiz J, Sevilla M-Á, Carrón R, and Montero M-J. Astaxanthin-enriched-diet reduces blood pressure and improves cardiovascular parameters in spontaneously hypertensive rats. Pharmacol Res 2011; 63:44–50. 

  1. ,,However, to the best of our knowledge, we have no reports about the prevention of diabetes in high-risk people, such as those with prediabetes with increased tissue oxidative stress and inflammation by the dietary intake of ASTX.” This sentence is not related to the study and needs to be correct.

10Reply: We changed as follows. 

However, to the best of our knowledge, we have no reports of human studies about the prevention of diabetes by the supplementation of ASTX in high-risk subjects, such as those with prediabetes with increased tissue oxidative stress and inflammation.

  1. Please provide the CONSORT checklist in the supplementary file.

11Reply: The CONSORT checklist in the supplementary file is necessary?

 If so, we need more time to provide the CONSORT checklist in the supplementary file. 

  1. Please explain how did you perform the randomisation?

12Reply: We mentioned the statement in Study population Section as follows. 

Randomization was performed by an assistant using block randomization method and stratified randomization was employed to match participants based on gender distribution. The intervention allocation was blinded for both investigators and participants.

  1. Where did the authors register the study protocol?

13Reply: We had not registered the study protocol.  The study protocol of this randomized clinical trial was approved by the Research Ethics Committee of the Japan Society of Nutrition and Food Science (IRB: H27-52). 

  1. Please provide more details about the methods which you used.

14Reply: We think that we have already described details about the methods.

Our revised manuscript is now checked by MDPI English Editing Services.

  1. Please check if the correct study is cited here: „Finally, Endo-PAT 2000 computer 164 software (Medical Itamar) was used to automatically collect signal data in order to obtain the reaction of vascular endothelial function of the vascular reactive hyperemia index (RHI) [15].”.

15Reply: We cite here the bellow report additionally. 

20.Naomi M. Hamburg and Emelia J. Benjamin.  Assessment of Endothelial Function Using Digital Pulse Amplitude Tonometry. Trends in Cardiovascular Medicine, 2009, 19, 6-11,

  1. Did the authors assess the dietary habits of the study population?

16Reply: We did not assess the dietary habits of the study population. 

  1. Did the authors calculate the minimal sample size?

17Reply: We did not calculate the minimal sample size. 

  1. In the Statistical analysis section please add the information on how you check the normal distribution of parameters?

18Reply: We checked the distribution of parameters by F-analysis.

  1. Please compare the effect of astaxanthin supplementation in normal and impaired fasting glucose subjects.

19Reply: We added the results of the effect of astaxanthin supplementation in in prediabetic subjects whose range of HbA1c is from 5.6 to 6.4, and changed to as follows.   

      The levels of HbA1c in all subjects was significantly decreased compared to that at baseline (5.64±0.33 vs 5.57±0.39ï¼…, *p < 0.05, Fig 4a), which is the first report in the human study. The levels of HbA1c in prediabetic subjects whose range of HbA1c is from 5.6 to 6.4 was also significantly decreased compared to that at baseline (5.82±0.22 vs 5.73±0.35ï¼…, *p < 0.05, Fig 4b). Please find attached.

  1. The authors should also compare the effect of astaxanthin supplementation between the astaxanthin and placebo groups.

20Reply: We add the below statement in Results section. 

We compared the changes in the level of HbA1c between the astaxanthin and placebo groups by Mann– Whitney U test.  No significant change was observed between the groups. 

  1. Are you sure that in Figure 1 the number of subjects who received the intervention should be 0?

21Reply: The number of subjects who received the intervention was not correct.  Figure 1 was corrected, please find attached. 

22.Participants were also asked to maintain their habitual diet, lifestyle, and tablet ingestion.” In the Study population section, you mentioned that participants were advised to maintain a healthy lifestyle.

22Reply: [participants were advised to maintain a healthy lifestyle.] was wrong.  We changed [participants were advised to maintain a healthy lifestyle] to [participants were advised to maintain their current lifestyle].  

23.Please add to Table 1 information about how many participants had prediabetes in both groups. 

23Reply: We added the number of prediabetes to Table 1 as below (attached).

BMI: body mass index,  BP: blood pressure,  n.s.; not significant, *p < 0.05. 

The Mann– Whitney U test was used to test for significant differences between the groups.

  1. The authors should add information about the difference between the groups at baseline.

24Reply: We add information about the difference between the groups at baseline in Table 1 as the above reply 22.

  1. The sentence should be check if it is correct: The levels of HbA1c 202 (5.64±33 vs 5.57±0.39ï¼…, p < 0.05, Fig 4), which is the first report in the human study..

25.Reply: We changed to [The levels of HbA1c in all subjects was significantly decreased compared to that at baseline (5.64±0.33 vs 5.57±0.39ï¼…, *p < 0.05, Fig 4a), which is the first report in the human study. The levels of HbA1c in prediabetic subjects whose range of HbA1c is from 5.6 to 6.4 was also significantly decreased compared to that at baseline (5.82±0.22 vs 5.73±0.35ï¼…, *p < 0.05, Fig 4b)].

  1. Please prepare Table 2 according to journal recommendation.

26.Reply: We changed Table 2 as follows. 

TC; Total cholesterol, Triglyceride, TG; Triglyceride

HDL-C; High-density lipoprotein-cholesterol, LDL; low-density lipoprotein,

MDA-LDL; Malondialdehyde-modified low-density lipoprotein,

   Our revised manuscript is now checked by MDPI English Editing Services. 

  1. Please explain how recommendations to provide a healthy lifestyle had an impact on the study results.

27.Reply: As mentioned in reply, [participants were advised to maintain a healthy lifestyle.] was wrong.  We changed [participants were advised to maintain a healthy lifestyle] to [participants were advised to maintain their current lifestyle].

  1. Please explain why the authors choose a dose 12 mg/d of astaxanthin?

28.Reply: We add the below statement in the method section.

      The daily ingestion amount of 12 mg and ingestion period of 12 weeks were set based on the previous reports [J Clin Biochem Nutr. 2009;44:280-4, **].

  1. Please check citations here: „In animal studies, several investigators reported that ASTX supplementation protected pancreatic beta-cells against glucose toxicity by decreasing the level of blood glucose and oxidative stress induced by hyperglycemia in diabetic db/db mice [12, 13, 16].

29.Reply: We insert the below report instead of 12, 13, and 16. 

Uchiyama K, Naito Y, Hasegawa G, Nakamura N, Takahashi J, Yoshikawa T.

Astaxanthin protects beta-cells against glucose toxicity in diabetic db/db mice.

Redox Rep. 2002;7:290-3.

  1. Please check the citation in the following sentence: „In concern with blood pressure, a decrease in blood pressure was also reported in patients receiving ASTX with type 2 diabetes mellitus [21]." The cited study was conducted in hypertensive rats.

30.Reply: We insert the below report instead of 21. 

24.Mashhadi NS, Zakerkish M, Mohammadiasl J, Zarei M, Mohammadshahi M, Haghighizadeh MH. Astaxanthin improves glucose metabolism and reduces blood pressure in patients with type 2 diabetes mellitus. Asia Pac J Clin Nutr. 2018;27:341-346.

  1. Please cite in the discussion the results of the meta-analysis which assessed the effect of astaxanthin supplementation on lipid profile and glucose levels.

31.Reply: We insert the below statements in the discussion. 

Ursoniu et al performed a meta-analysis to evaluate the efficacy of astaxanthin supplementation on plasma lipid and glucose concentrations, and reported that a slight glucose-lowering effect, not HbA1c, was observed.

Ursoniu et al reported by the meta-analysis that a significant effect of supplementation with astaxanthin on plasma lipid profile was not observed.

21.Sorin Ursoniu, Amirhossein Sahebkar, Maria-Corina Serban, Maciej Banach.

Lipid profile and glucose changes after supplementation with astaxanthin:

a systematic review and meta-analysis of randomized controlled trials

Arch Med Sci. 2015;11:253-66.

  1. What are the strengths and novelty of the study?

32.Reply: The strengths and novelty of the study is that the level of HbA1c was reduced by ASTX supplementation in healthy volunteers and subjects with prediabetes.

  1. Please add the conclusions section to the paper.

33.Reply: W add the conclusions section as follows. 

CONCLUSION

We demonstrated that ASTX decreases the level of HbA1c in healthy volunteers and subjects with prediabetes.  Our results suggest that ASTX may play a role in glucose control and that may be clinically beneficial for prevention of type 2 diabetes in subjects with or without prediabetes. Furthermore, our results indicate that ASTX may be a novel complementary treatment with potential impacts on diabetes and diabetic complications without adverse effects.

Reviewer 2 Report

Hi, Please see attached.

Author Response

Reviewer Comments

This study aims to identify the potential of astaxanthin compound in humans particularly its glucose lowering effects and effects on endothelial function. ASTX has primarily been studied to reduce oxidative stress and has not been tested in the context of diabetes. Thus, the authors chose healthy volunteers to see if ASTX regulated glucose metabolism. Results showed that ASTX regulated glucose metabolism and also endothelial function demonstrating its importance in context of atherosclerosis. The authors have taken extra care in writing the method section in a very detailed manner. A good detailed overview of the study population and study protocol helps to understand the study. There are some aspects of the manuscript that authors need to work on.

Some of my comments/feedback are provided below.

Major comments

  • Astaxanthin was proved to humans at a dose of 12 mg daily for 12 weeks. It is not clear as to how did the authors conclude to use this dose of ASTX for this amount of time. There is no indication to any references. It is thus suggested that authors provide a brief explanation in their method section as to how they came up with this dose etc.

Reply: We add the below statement in the method section.

      The daily ingestion amount of 12 mg and ingestion period of 12 weeks were set based on the previous reports [J Clin Biochem Nutr. 2009;44:280-4, **].

Akira Satoh 1 , Shinji Tsuji, Yumika Okada, Nagisa Murakami, Maki Urami, Keisuke Nakagawa, Masaharu Ishikura, Mikiyuki Katagiri, Yoshihiko Koga, Takuji Shirasawa. Preliminary Clinical Evaluation of Toxicity and Efficacy of A New Astaxanthin-rich Haematococcus pluvialis Extract. J Clin Biochem Nutr. 2009;44:280-4.

  • The authors have emphasised in the title and elsewhere that prediabetic patients were included in the trial along with healthy volunteers. It would have been better to state that if they found any differences in the levels of HbA1c in the prediabetic patients too. Did the authors do a separate analysis to see if ASTX had an impact on just the prediabetic population?

Reply: We changed the results of [Changes in the glucose metabolic parameters] as below;

      The levels of HbA1c in all subjects was significantly decreased compared to that at baseline (5.64±0.33 vs 5.57±0.39ï¼…, *p < 0.05, Fig 4a), which is the first report in the human study. The levels of HbA1c in prediabetic subjects whose range of HbA1c is from 5.6 to 6.4 was also significantly decreased compared to that at baseline (5.82±0.22 vs 5.73±0.35ï¼…, *p < 0.05, Fig 4b), attached. 

  • In figure 3, a significance indicated by asterisk has also been found in the placebo group after 75g-OGTT test but no further explanation has been provided in the manuscript. Please provide possible explanation in the discussion section.

Reply: It is very hard to provide explanation about the significant difference at 60min after 75g-OGTT in the placebo group. It might be one of placebo effects, however, we do not know.  We think it is not a scientific comment. Please forgive us to mention no appropriate explanation in the discussion section.      

Minor comments

  • At various places in the manuscript, there are typos that needs to be corrected. For instance Line # 20 recently is spelled in correct. This is just one of the many typos. It is recommended that authors go through the manuscript thoroughly.

Reply: [Recnely] was changed to [Recently].

           Our revised manuscript is now checked by MDPI English Editing Services.

  • Lines #29-30 needs to be paraphrased to avoid any grammatical mistakes. At the moment, it does not read well.

Reply:   Our revised manuscript is now checked by MDPI English Editing Services.

  • Consistency of 75 g-OGTT should be looked into. Example in Line #30vs Line#39 vs Line #52. Please be consistent throughout

Reply:   Our revised manuscript is now checked by MDPI English Editing Services.

  • Line#80- The last part of the sentence (which is link to the…..antioxidants) is grammatically incorrect

Reply:   Our revised manuscript is now checked by MDPI English Editing Services.

  • In Figure 2, x-axis is in months (M) but the elsewhere the text says as 0, 4, 8 and 12 weeks. Please change the x-axis to weeks also, if you have been using weeks in the manuscript. Please stick to one type description for time for an experiment.

Reply: We changed [M] to [W]. Attached.

  • In Line 246-248, it is said that Mashadi et al has done a study using ASTX supplementation and diabetic patients but no reference is provided. Please double check and provide a reference.

Reply; We provided the reference, Mashhadi NS, Zakerkish M, Mohammadiasl J, Zarei M, Mohammadshahi M, Haghighizadeh MH. Astaxanthin improves glucose metabolism and reduces blood pressure in patients with type 2 diabetes mellitus. Asia Pac J Clin Nutr. 2018;27:341-346.

Our revised manuscript is now checked by MDPI English Editing Services.

  • Line # 266 and 270. - In concern with…seems redundant and repetitive. Please paraphrase

Reply:   Our revised manuscript is now checked by MDPI English Editing Services.

  • Line # 271 – WANG in capitals – why is it like that?

Reply:  WANG in capitals changed to Wang.

Our revised manuscript is now checked by MDPI English Editing Services.

Round 2

Reviewer 1 Report

Unfortunately, the manuscript was not significantly improved. Please find below several comments.

General comments:
Not all changes were marked in the manuscript.
Please prepare the manuscript according to the CONSORT guidelines.

Title:
Identification as a randomised trial should be included in the title.

Abstract:
Please add the information about differences between groups.

Aim of the study:
Please change "healthy volunteers, including subjects with prediabetes" to "healthy volunteers and subjects with prediabetes".

Methods:
If your study was conducted according to the CONSORT guidelines you should include the CONSORT checklist in the Supplementary File.

Randomisation:
Stratification should also include subjects' healthy status. Please add this information to the study limitation.

Statistic analysis:
Please provide the information if the data were normal distributed? If not, the data should also be presented using the median and interquartile range.

Results:
Figure 1 is unreadable.
How do the authors explain no differences in changes in the level of HbA1c between groups? According to the results, ASTX was not more effective than placebo in decrease HbA1c. Therefore, please consider changing the conclusion of the study.
Please provide a separate table with the comparison of the changes in the analysing parameters between groups.
Was the ASTX more effective in the prediabetes group? Please add a short sum-up of the results.

Discussion:
Please add the strengths of the study to the manuscript.
Lack of assessment of dietary habits should be added to the limitation.